# LEARN TO LEARN CONSISTENTLY

## ABSTRACT

In the few-shot learning problem, a model trained on a disjoint meta-train dataset is required to address novel tasks with limited novel examples. A key challenge in few-shot learning is the model's propensity to learn biased shortcut features(e.g., background, noise, shape, color), which are sufficient to distinguish the few examples during fast adaptation but lead to poor generalization. In our work, we observed when the model learns with higher consistency, the model tends to be less influenced by shortcut features, resulting in better generalization. Based on the observation, we propose a simple yet effective meta-learning method named Meta Self-Distillation. By maximizing the consistency of the learned knowledge during the meta-train phase, the model initialized by our method shows better generalization in the meta-test phase. Extensive experiments demonstrate that our method improves the model's generalization across various few-shot classification scenarios and enhances the model's ability to learn consistently.

## 1 INTRODUCTION

Few-shot learning aims to address novel tasks with a limited number of examples, typically through rapid adaptation of a model trained on a dataset with disjoint labels. Many approaches tackle this issue from the perspective of meta-learning(Finn et al., 2017; Lee et al., 2019; Ravi & Larochelle, 2016; Lake & Baroni, 2023). Methods such as Model-Agnostic meta-Learning (MAML) (Finn et al., 2017) and its variants (Raghu et al., 2019; Ye & Chao, 2021; Antoniou et al., 2018; Kao et al., 2021; Nichol et al., 2018) aim to learn initialized parameters for a model with prior knowledge for fast adaptation. Recent research has explored more challenging scenarios, such as cross-domain few-shot learning (Ullah et al., 2022; Triantafillou et al., 2019; Tseng et al., 2020; Guo et al., 2020), where the novel task belongs to a different domain and label set than the training dataset.

A key challenge in various few-shot learning problems is the model's tendency to learn biased shortcut features (e.g., background, noise, shape, color) from limited examples (Shah et al., 2020; Teney et al., 2022; Lyu et al., 2021; Le et al., 2021). These shortcut features may suffice to distinguish the few classes during rapid adaptation but result in poor generalization. Several solutions have been proposed to address these issues. Although these approaches partially mitigate the problem, they often require additional resources or learn generalized features only within the meta-train dataset (Le et al., 2021; Zhou et al., 2023; Liu et al., 2020; Dvornik et al., 2020; Snell et al., 2017). From the perspective of meta-learning, we ask the following question: *Can we make the initialized model more inclined to learn generalization features rather than shortcut features when addressing novel tasks?*

This problem is challenging to address directly, as identifying generalized versus shortcut features in the data is difficult. In our study, we generate different views of the same data through data augmentation, which makes these views have different shortcut features but similar generalized features. We use these views to update the model and observe that when model learning with better consistency tends to exhibit better generalization. This implies that when tasks are learned with higher consistency by the model, the model is less influenced by the shortcut features and reaches higher accuracy. At this point, if we can enhance the model's consistency of learning across all tasks, we can make the model less influenced by the shortcut feature and more inclined to learn generalized features.

Based on this observation and inspired by the idea of self-distillation (Caron et al., 2020; Chen & He, 2021; Caron et al., 2021), we proposed meta self-distillation, which aims to maximize the

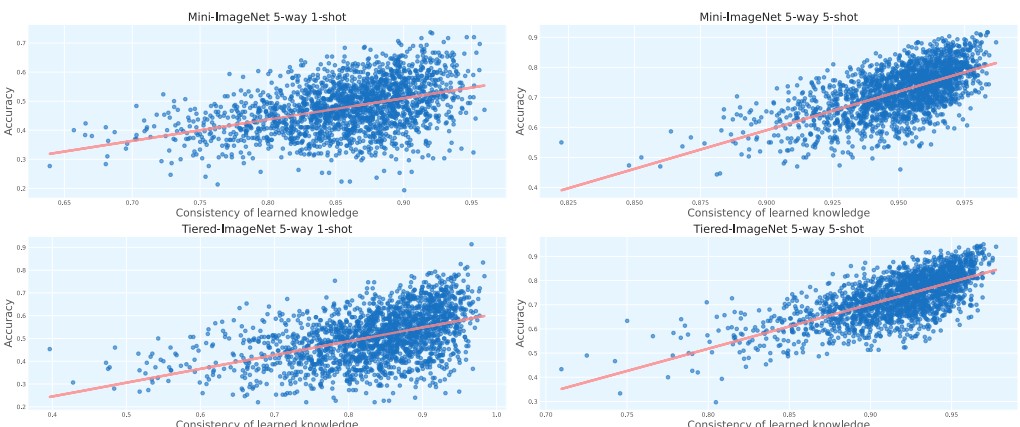

Figure 1: **The core idea between self-distillation and meta self-distillation.** Self-distillation aims to make the deep representation of different views closer, while meta self-distillation aims to learn consistent knowledge from the different views of the same image.

Figure 2: **The consistency versus accuracy of the model initialized by MAML across different tasks.** The results demonstrate a clear trend: as the model's consistency learned from the task increases, the average accuracy in predicting query data improves.

consistency of models updated from the initialized model by using augmented views of the same data. Specifically, in the inner loop, we augment the same tasks to update the initialized model independently. In the outer loop, we maximize the consistency of the outputs for the same query data produced by the differently updated models. This approach enhances the initialized model's ability to learn consistently, thereby improving the generalization of the initialized model. We evaluate our method in three different settings of few-shot learning, which has demonstrated the effectiveness of our method.

In summary, our contributions are as follows:

- We observed that the consistency of model learning could serve as an indicator of the model's inclination towards learning shortcut features that lead to overfitting.

- We proposed a meta-learning method named meta self-distillation(MSD). By maximizing the consistency of learned knowledge, MSD improves the initialized model's ability to learn more consistently, thereby making the model more inclined to learn generalized features.

- Extensive experiments demonstrate that our method achieves remarkable performance across various few-shot scenarios and significantly enhances the model's ability to learn consistently in unseen tasks.

## 2 RELATED WORK

### 2.1 FEW-SHOT LEARNING

Few-shot learning aims to address novel tasks with a limited number of examples, typically through the rapid adaptation of a model trained on base classes, which are disjoint from the classes in novel tasks. Solutions to few-shot learning are primarily categorized into meta-learning and transfer learn-

ing approaches. meta-learning (Antoniou et al., 2018; Finn et al., 2017; Ye & Chao, 2021) aims to train a model with prior knowledge that can fast adapt to novel tasks. Transfer learning (Tian et al., 2020; Mangla et al., 2020; Liu et al., 2021) focuses on developing a generalized feature extractor from base classes that can generalize to novel tasks. Traditionally, few-shot learning assumes that base and novel classes originate from the same domain but differ in categories. Recent studies have extended this to cross-domain few-shot learning, where base and novel classes belong to different domains (Ullah et al., 2022; Triantafillou et al., 2019; Tseng et al., 2020; Guo et al., 2020). A critical challenge in few-shot learning is that during fast adaptation, models tend to learn shortcut features, leading to overfitting on novel tasks (Shah et al., 2020; Teney et al., 2022; Lyu et al., 2021; Le et al., 2021). Various methods have been proposed to address this issue. For instance, Poodle (Le et al., 2021) suggests using additional data to penalize out-of-distribution samples, while LDP-net (Zhou et al., 2023) employs local and global knowledge distillation to enable the model to learn more diverse features from the meta-training dataset. Although these methods mitigate the problem to some extent, they often require additional data or parameters to adapt to unseen tasks and domains. Alternatively, some approaches train a powerful feature extractor solely on the meta-train dataset, which may limit the model's ability to recognize unseen features in novel tasks (Le et al., 2021; Zhou et al., 2023; Liu et al., 2020; Dvornik et al., 2020; Snell et al., 2017). Our method aims to make the initialized model inclined to learn generalized features, thereby avoiding such limitations.

## 2.2 META-LEARNING

Meta-learning, also known as learning to learn, aims to learn initialized parameters with prior knowledge for fast adaptation. It is mainly divided into metric-based meta-learning, represented by ProtoNet (Snell et al., 2017), and optimize-based meta-learning, represented by MAML (Finn et al., 2017). Metric-based meta-learning improves model representation by bringing the representation between the support data and the query data that belong to the same category closer, typically not requiring fine-tuning during the meta-test phase. Optimize-based meta-learning aims to provide the initial parameters with prior knowledge, offering better generalization performance when fine-tuning on novel category samples. This category includes algorithms like MAML (Finn et al., 2017) and its variants, such as (Ye & Chao, 2021), which utilizes a single vector to replace the network's classification head weight, thus preventing the permutation in the meta-test phase. MAML++ (Antoniou et al., 2018) enhances MAML's performance by addressing multiple optimization issues encountered by MAML, while ANIL (Raghu et al., 2019) improves MAML's performance by freezing the backbone during the inner loop. In our work, we mainly focus on the optimize-based meta-learning. From the perspective of meta-learning, our goal is to train initialized parameters that incline to learn generalized features rather than shortcut features, thereby enhancing accuracy in the few-shot learning problems.

## 2.3 SELF-DISTILLATION

Self-distillation is a variant of contrastive learning (Caron et al., 2020; Chen & He, 2021), which is trained by bringing the representations of positive instance pairs closer without using negative pairs. BYOL (Grill et al., 2020)utilizes the exponential moving average of the network to produce the target of an online network. SimSiam (Chen & He, 2021) further explored how self-distillation avoids collapse in a self-supervised setting. (Allen-Zhu & Li, 2020) suggests that self-distillation can serve as an implicit ensemble distillation, allowing the model to distinguish more view features. Self-distillation is an effective method to enhance the model's feature extraction capabilities and can be combined with meta-learning (Li et al., 2022; Ni et al., 2021). Typically, self-distillation aims to maximize the similarity of the representations across different views. Different from the typical self-distillation that directly aligns the representation, we propose to use meta self-distillation to maximize the consistency of the different updated models' outputs. In this way, we can make the initialized model less influenced by shortcut features when addressing a new task.

## 3 PRELIMINARY

Here, we provide an overview of the fundamental setting and problem for few-shot learning classification, along with an introduction to model-agnostic meta-learning (MAML).

### 3.1 PROBLEM DEFINITION FOR FEW-SHOT CLASSIFICATION

Following (Vinyals et al., 2016; Chen et al., 2019; Wang et al., 2020), We define the few-shot classification problem(FSL) as an $\mathcal{N}$-way $\mathcal{K}$-shot task, where there are $\mathcal{N}$ classes, each containing $\mathcal{K}$-labeled support samples. Typically, $\mathcal{K}$ is small, such as 1 or 5. The data used to attempt to update the model is defined as the support data $\mathcal{S} = \{x_s, y_s\}$, where each $x_s$ represents the model's input, and $y_s$ denotes the corresponding label for $x_s$. The data used to evaluate the effectiveness of the model updates is defined as the query data $\mathcal{Q} = \{x_q, y_q\}$, which has the same class as the support data, but the samples contained in the query set are different from those in the support set. The FSL task is defined as the problem of learning to correctly classify the query data $\mathcal{Q}$ with the support data $\mathcal{S}$, which can be written as follows:

$$\arg\min_{\theta} \mathbb{E}_{\mathcal{S},\mathcal{Q}\sim\mathcal{D}_{\text{meta-test}}} \left[ \mathcal{L}_{\text{FSL}}(\theta, \mathcal{S}, \mathcal{Q}) \right] \tag{1}$$

If the model is randomly initialized and directly fine-tuned on the limited support data, the model will overfit. To address that, we need to transfer knowledge from seen data to the unseen data. The seen data used in FSL is referred to as the meta-train set, and the unseen data is referred to as the meta-test set. The labels in the two sets are disjoint, and in the cross-domain few-shot learning, the domains of the two sets are also different. The goal of FSL is to pretrain or initialize the parameters by using the meta-train set and generalize to the unseen task sampled from the meta-test set.

### 3.2 MODEL-AGNOSTIC META-LEARNING

Model Agnostic Meta-Learning (MAML) (Finn et al., 2017) is a meta-learning framework. The objective of MAML is to learn initialized parameters $\theta$ with prior knowledge, such that after a few steps of standard training on the support data, the model can generalize well on the query data. The objective can be as follows:

$$\arg\min_{\theta} \mathbb{E}_{\mathcal{S},\mathcal{Q}\sim\mathcal{D}_{\text{meta-train}}} \left[ \mathcal{L}(\mathcal{U}^k(\theta, \mathcal{S}), \mathcal{Q}) \right] \tag{2}$$

Where $\mathcal{U}^k$ denotes $k$ updates of the parameter $\theta$ using tasks sampled from the task distribution, which corresponds to adding a sequence of gradient vectors to the initialized parameters:

$$\mathcal{U}^k(\theta, \mathcal{S}) = \theta - \sum_{i=1}^{k} \alpha \cdot \frac{\partial \mathcal{L}(\mathcal{U}^{i-1}(\theta, \mathcal{S}), \mathcal{S})}{\partial \theta}, \quad \mathcal{U}^0(\theta, \mathcal{S}) = \theta \tag{3}$$

The process of updating the parameters with support data is referred to as *inner loop process*, where $\alpha$ is the stepsize of the inner loop. Subsequently, the query data $\mathcal{Q}$ is used to evaluate $\mathcal{U}^k(\theta, \mathcal{S})$, and directly updating the initial parameters $\theta$, which known as the *outer loop process*. The outer loop commonly employs SGD for updates, and the update process can be computed as follows:

$$\theta' = \theta - \beta \cdot \frac{\partial \mathcal{L}(\mathcal{U}^k(\theta, \mathcal{Q}), \mathcal{S})}{\partial \theta} \tag{4}$$

Where $\beta$ is the learning rate of the outer loop. By minimizing the loss across sampled tasks, MAML enables the parameters to learn prior knowledge from the meta-train set.

## 4 LEARN TO LEARN CONSISTENTLY

### 4.1 WHY LEARN CONSISTENTLY IN FSL

Previous studies have indicated that in few-shot learning (FSL) scenarios, models tend to learn shortcut features (e.g., background, noise, shape, color) from limited examples (Shah et al., 2020; Teney et al., 2022; Lyu et al., 2021; Le et al., 2021). These shortcut features may suffice to distinguish the few classes during rapid adaptation but often lead to poor generalization. From the perspective

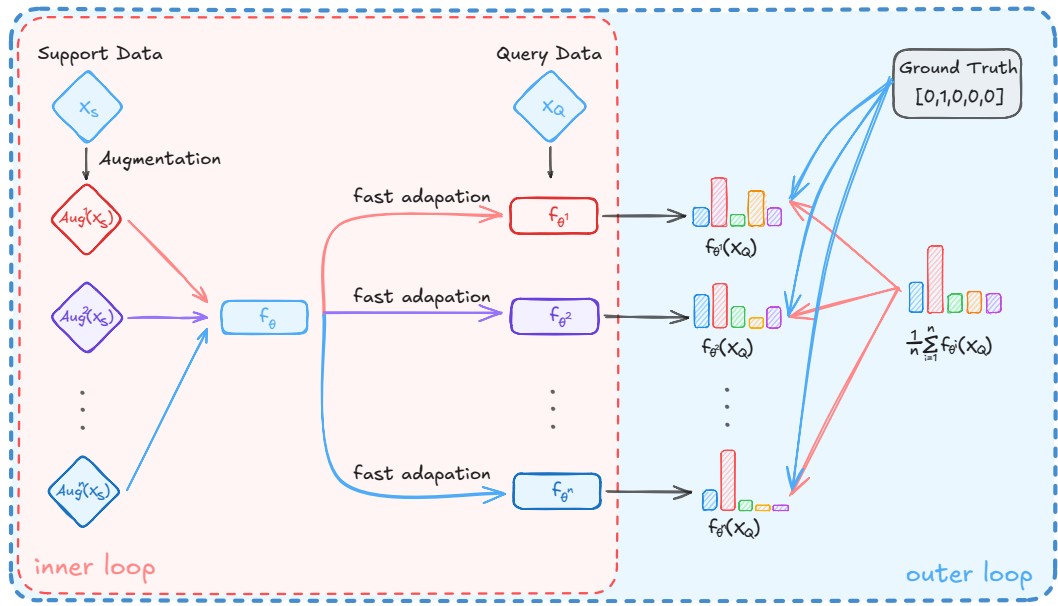

Figure 3: **An overview of the proposed MSD.** In the inner loop, MSD first uses different augmented support data to update the $f_\theta$. In the outer loop, then maximizes the consistency among the outputs of the same query data with different update versions of the initial model

---

**Algorithm 1:** Evaluate the consistency and accuracy of model initialized by MAML

**Given** the learned initialization $\theta$ by MAML
**for** $t \in \{1, \cdots, T\}$ **do**
    **Sample** task $\mathcal{T} = (\mathcal{S}, \mathcal{Q}) \sim \mathcal{D}_{\text{meta-test}}$
    **for** $i \in \{1, \cdots, n\}$ **do**
        **Random Augmented** the support data: Get $\mathcal{S}_i = \mathcal{A}ug(\mathcal{S})$
        **Update** $\theta$ by augmented support data $\mathcal{S}_i$: Get $\theta_i = \mathcal{U}^k(\theta, \mathcal{S}_i)$
        **Get the output** of the query data $x_q$ by $\theta_i$: Get $v_i = f_{\theta_i}(x_q)$
    **end**
    **Record** the consistency and average accuracy of the output $\{v_i\}$:
        $\mathcal{C}[t] = \frac{1}{n}\sum_{i=1}^n \mathcal{F}_{\text{sim}}\left(v_i, \frac{1}{n}\sum_{i=1}^n v_i\right), \mathcal{A}[t] = \frac{1}{n}\sum_{i=1}^n \mathcal{A}cc(v_i)$
**end**
**Return** $\mathcal{C}, \mathcal{A}$

---

of meta-learning, we aim for the initialized model to learn more generalized features, avoiding the reliance on shortcut features. However, it's hard to distinguish these features directly in practice. To solve that, We proposed that one can make the initialized model less influenced by the shortcut features by enhancing the model's consistency in learning.

To validate the point, we evaluate the consistency and accuracy of the parameters initialized by MAML. During the meta-test phase, we sample one task and augment its support data, updating the initialized parameters respectively. To evaluate the knowledge acquired from these support data, we tested the updated models using the same query data and recorded the average prediction accuracy and output consistency across the tasks. For each task group, the support data are augmented from the same data, thus containing different shortcut features and similar generalized features. Therefore, the inconsistency of the differently updated models is mainly caused by different shortcut features. When the model is more inclined to learn generalized features, its outputs for the same query data should be similar. In contrast, if the model tends to learn shortcut features, which results in overfitting, the augmented inconsistencies in these features lead to greater output variance for the same query data. The results are illustrated in Figure 2: lower consistency corresponds to lower average prediction accuracy. Additionally, as the amount of support data increases, the model is less influenced by shortcut features, and both consistency and accuracy are improved consistently. Consistency and accuracy exhibit a high degree of alignment In various settings. Therefore, we demonstrate that model consistency reflects the tendency to learn shortcut features. Based on the

observation, We propose to enhance the consistency of learning across all tasks to make the initialized model less influenced by the shortcut feature during fast adaptation. Therefore, we define the objective of "learn to learn consistently" as follows:

$$\arg\max_{\theta} \mathbb{E}_{\mathcal{S},\mathcal{Q} \sim \mathcal{D}_{\text{meta-test}}} \left[ \mathcal{F}_{\text{sim}} \left( v_i, \bar{v} \right) \right] \tag{5}$$

Where $\mathcal{F}_{\text{sim}}$ is the similar function to evaluate consistency, $v_i$ is the output of sampled query data by different updated models. When $\mathcal{F}_{\text{sim}}$ is the negative mean squared error, the objective is to minimize the variance of the output among the different updated models.

## 4.2 META SELF-DISTILLATION

Based on the objective proposed in Eq.5, we propose a meta-learning method named meta self-distillation to enhance the model's ability to learn consistently.

**Meta-Train Phase.** Specifically, we sample tasks from the meta-train set to obtain support and query data. Unlike MAML, which samples multiple tasks, we sample a single task and create multiple augmented versions as substitutes. Only the support data is augmented in the different augmented tasks, and the tasks share the same query data. The rationale behind this is to have the same standard when assessing the knowledge learned by the model. Let the tasks be denoted as $\mathcal{T} = \{\mathcal{S}_i, \mathcal{Q}\}$, where $\mathcal{S}_i$ represent the $i$-th augmented view of support data. In the inner loop, we update the model with different augmented views of the support data to obtain varied models:

$$\theta_i = \mathcal{U}^k(\theta, \mathcal{S}_i) = \theta - \sum_{i=1}^{k} \alpha \cdot \frac{\partial \mathcal{L}(\mathcal{U}^{k-1}(\theta, \mathcal{S}_i), \mathcal{S}_i)}{\partial \theta}, \quad \mathcal{U}^0(\theta, \mathcal{S}_i) = \theta \tag{6}$$

In the outer loop, we test the query with different updated versions of the parameters. Since we desire the model to extract the same knowledge from different augmented views of support data, we measure the consistency of their query outputs to assess if the knowledge learned is identical:

$$\mathcal{L}_{\text{CK}} = -\frac{1}{n} \sum_{i=1}^{n} \mathcal{F}_{\text{sim}} \left( f_{\theta_i}(x_q), \frac{1}{n} \sum_{i=1}^{n} f_{\theta_i}(x_q) \right) \tag{7}$$

Where $\mathcal{F}_{\text{sim}}$ is the similarity function. Following (Chen & He, 2021), we use cosine similarity as the similarity function in practice. Furthermore, to ensure the model fully utilizes label information and learns precise classification, we also compute the classification loss for each updated parameter by query data. The model's total loss is expressed as:

$$\mathcal{L}_{\text{total}} = \mathcal{L}_{\text{cls}} + \gamma \cdot \mathcal{L}_{\text{CK}} \tag{8}$$

Where $\gamma$ represents the coefficient of consistency loss. The process of updating the initial parameters is as follows:

$$\theta' = \theta - \beta \cdot \nabla_{\theta} \mathcal{L}_{\text{total}} \tag{9}$$

Where $\beta$ represents the learning rate in the outer loop. The specific process has been shown in Algorithm 2 in the Appendix.

**Meta-Test Phase.** During the meta-test phase, MSD is consistent with MAML. We perform fast adaptation on the input support data using SGD and classify the query data directly with the updated model.

## 5 EXPERIMENT

### 5.1 EXPERIMENT SETTING

**Datasets.** For standard and augmented FSL evaluation, Our method was primarily evaluated on two benchmark datasets: Mini-ImageNet (Vinyals et al., 2016) and Tiered-ImageNet (Ren et al., 2018), both widely used for few-shot learning assessments. For cross-domain FSL evaluation, we use Mini-ImageNet as the source domain and use another eight datasets as the target domain, i.e., CUB, Cars, Places, Plantae, ChestX, ISIC, EuroSAT and CropDisease.

Table 1: **5way-1shot and 5way-5shot classification accuracy in standard few-shot classification task** and 95% confidence interval on Mini-ImageNet and Tiered-ImageNet (over 2000 tasks), using ResNet-12 as the backbone. NIW-Meta used ResNet-18 as the backbone.

| Methods | Mini-ImageNet | | Tiered-ImageNet | |
|---|---|---|---|---|
| | 1-Shot | 5-Shot | 1-Shot | 5-Shot |
| ProtoNet (Snell et al., 2017) | $62.39 \pm 0.20$ | $80.53 \pm 0.20$ | $68.23 \pm 0.23$ | $84.03 \pm 0.16$ |
| MAML (Finn et al., 2017) | $64.42 \pm 0.20$ | $83.44 \pm 0.14$ | $65.72 \pm 0.20$ | $84.37 \pm 0.16$ |
| MetaOptNet (Lee et al., 2019) | $62.64 \pm 0.35$ | $78.63 \pm 0.68$ | $65.99 \pm 0.72$ | $81.56 \pm 0.53$ |
| ProtoMAML (Triantafillou et al., 2019) | $64.12 \pm 0.20$ | $81.24 \pm 0.20$ | $68.46 \pm 0.23$ | $84.67 \pm 0.16$ |
| DSN-MR (Simon et al., 2020) | $64.60 \pm 0.72$ | $79.51 \pm 0.50$ | $67.39 \pm 0.82$ | $82.85 \pm 0.56$ |
| Meta-AdaM (Sun & Gao, 2024) | $59.89 \pm 0.49$ | $77.92 \pm 0.43$ | $65.31 \pm 0.48$ | $85.24 \pm 0.35$ |
| LA-PID (Yu et al., 2024) | $63.29 \pm 0.48$ | $79.18 \pm 0.43$ | $64.77 \pm 0.47$ | $82.59 \pm 0.37$ |
| NIW-Meta[†] (Kim & Hospedales, 2024) | $65.49 \pm 0.56$ | $81.71 \pm 0.17$ | $70.52 \pm 0.19$ | $85.83 \pm 0.17$ |
| MSD | $\mathbf{65.41 \pm 0.47}$ | $\mathbf{84.88 \pm 0.29}$ | $68.51 \pm 0.53$ | $\mathbf{86.87 \pm 0.34}$ |

Table 2: **5way-5shot classification accuracy in cross-domain few-shot classification task** (over 2000 tasks), using ResNet-12 as the backbone. Only the meta-train set of Mini-ImageNet is used during the meta-train phase.

| | CUB | Cars | Places | Plantae | Euro | ISIC | CropD | ChestX |
|---|---|---|---|---|---|---|---|---|
| GNN (Garcia & Bruna, 2017) | 62.87 | 43.70 | 70.91 | 48.51 | 78.69 | 42.54 | 83.12 | 23.87 |
| GNN+FT (Tseng et al., 2020) | 64.97 | 46.19 | 70.70 | 49.66 | 78.02 | 40.87 | 87.07 | 24.28 |
| TPN+ATA (Wang & Deng, 2021) | 70.14 | 55.23 | 73.87 | 59.02 | 85.47 | 49.83 | 93.56 | 24.74 |
| GNN+ATA (Wang & Deng, 2021) | 66.22 | 49.14 | 75.48 | 52.69 | 83.75 | 44.91 | 90.59 | 24.32 |
| MatchingNet+AFA (Hu & Ma, 2022) | 59.46 | 46.13 | 68.87 | 52.43 | 69.63 | 39.88 | 80.07 | 23.18 |
| GNN+AFA (Hu & Ma, 2022) | 68.25 | 49.28 | **76.21** | 54.26 | 85.58 | 46.01 | 88.06 | 25.02 |
| LDP-net (Zhou et al., 2023) | **70.39** | 52.84 | 72.90 | 58.49 | 82.01 | 48.06 | 89.40 | 26.67 |
| GNN +FAP (Zhang et al., 2024) | 67.66 | 50.20 | 74.98 | 54.54 | 82.52 | 47.60 | 91.79 | 25.31 |
| RFS+MLP (Bai et al., 2024) | - | - | - | - | 83.14 | 46.02 | 66.87 | **29.09** |
| MSD | 70.22 | **58.55** | 75.59 | **60.81** | **85.65** | **51.54** | **95.12** | 28.26 |

The Mini-ImageNet dataset comprises 100 classes, each containing 600 samples. Following prior work, we divided the 100 classes into training, validation, and test sets, containing 64, 16, and 20 classes, respectively. The Tiered-ImageNet dataset encompasses 608 fine-grained classes, which are categorized into 34 higher-level classes. In alignment with previous studies, we divided these higher-level classes into training, validation, and test sets, comprising 20, 6, and 8 higher-level classes, respectively. Tiered-ImageNet is designed to consider class similarity when segmenting the dataset, ensuring a significant distributional difference between training and test data. CUB, Cars, Places, and Plantae proposed in (Tseng et al., 2020) contain natural images of different properties. ChestX, ISIC, EuroSAT and CropDisease proposed in (Guo et al., 2020) are cross-domain datasets from the domain of medicine, agriculture, and remote sensing, which have significant domain shifts. All the images are resized to $84 \times 84$ pixels following common practice.

**Backbone Model.** For our model evaluation, following (Lee et al., 2019), we employed a ResNet-12 (He et al., 2016) architecture, noted for its broader widths and Dropblock modules as introduced by (Ghiasi et al., 2018). This backbone is broadly used across numerous few-shot learning algorithms. Additionally, we follow the original MAML approach, utilizing a 4-layer convolutional neural network(Conv4) (Vinyals et al., 2016). Following the recent practice (Ye et al., 2020; Qiao et al., 2018; Rusu et al., 2018), The models' weights are pre-trained on the meta-train set to initialize.

**Experiment Details.** The other details are listed in the Appendix A.1

## 5.2 RESULTS

We evaluate our method under three settings: standard few-shot learning problems, cross-domain few-shot learning problems, and augmented few-shot learning problems.

Table 3: **5way-1shot and 5way-5shot classification accuracy** in augmented few-shot classification task and 95% confidence interval on Mini-ImageNet and Tiered-ImageNet (over 2000 tasks), using Conv4 as the backbone.the terms "strong" and "weak" denote the varying levels of augmentation applied to the support data in the meta-test phase.

| | | Mini-ImageNet (Strong) | | Mini-ImageNet (Weak) | |
|---|---|---|---|---|---|
| Methods | Backbone | 1-Shot | 5-Shot | 1-Shot | 5-Shot |
| MAML | Conv4 | $28.13 \pm 0.29$ | $37.77 \pm 0.31$ | $35.89 \pm 0.35$ | $49.54 \pm 0.36$ |
| MSD + MAML | Conv4 | $\mathbf{30.64 \pm 0.30}$ | $\mathbf{40.79 \pm 0.33}$ | $\mathbf{37.11 \pm 0.37}$ | $\mathbf{50.38 \pm 0.37}$ |
| Unicorn-MAML | Conv4 | $29.26 \pm 0.30$ | $40.58 \pm 0.33$ | $36.07 \pm 0.36$ | $51.43 \pm 0.37$ |
| MSD + Unicorn-MAML | Conv4 | $\mathbf{31.37 \pm 0.32}$ | $\mathbf{42.59 \pm 0.33}$ | $\mathbf{38.94 \pm 0.38}$ | $\mathbf{54.11 \pm 0.37}$ |

Table 4: **5way-1shot and 5way-5shot classification accuracy** in strongly augmented few-shot classification task and 95% confidence interval on Mini-ImageNet and Tiered-ImageNet (over 2000 tasks), using ResNet-12 as the backbone.

| | | Mini-ImageNet | | Tiered-ImageNet | |
|---|---|---|---|---|---|
| Methods | Backbone | 1-Shot | 5-Shot | 1-Shot | 5-Shot |
| MAML | ResNet-12 | $49.94 \pm 0.43$ | $73.46 \pm 0.36$ | $51.87 \pm 0.48$ | $75.11 \pm 0.39$ |
| MSD + MAML | ResNet-12 | $\mathbf{57.31 \pm 0.44}$ | $\mathbf{78.32 \pm 0.33}$ | $\mathbf{55.79 \pm 0.49}$ | $\mathbf{76.49 \pm 0.39}$ |
| Unicorn-MAML | ResNet-12 | $50.57 \pm 0.43$ | $73.68 \pm 0.35$ | $53.01 \pm 0.49$ | $76.08 \pm 0.40$ |
| MSD + Unicorn-MAML | ResNet-12 | $\mathbf{57.75 \pm 0.44}$ | $\mathbf{77.25 \pm 0.33}$ | $\mathbf{56.39 \pm 0.47}$ | $\mathbf{78.11 \pm 0.38}$ |

### 5.2.1 STANDARD FEW-SHOT LEARNING PROBLEMS.

The results in Table.10 demonstrate the performance of MSD and several mainstream few-shot algorithms on few-shot tasks. MSD exhibits a significant improvement over MAML in standard few-shot tasks. The results of maml are produced by(Ye & Chao, 2021), which uses more inner steps for maml to reach better performance. On Mini-ImageNet, our method achieved an increase of 0.99% in 5way-1shot and 1.44% in 5way-5shot tasks compared with maml, respectively. On Tiered-ImageNet, the improvements for 5way-1shot and 5way-5shot tasks were 2.79% and 2.50% compared with MAML, respectively. MSD shows excellent effectiveness in few-shot tasks, with better performance compared to the recent meta-learning algorithms and MAML's variants.

### 5.2.2 CROSS DOMAIN FEW-SHOT LEARNING PROBLEMS.

To explore the performance when there is a large domain gap between the meta-train set and the meta-test set, we also evaluated the performance of MSD under the cross-domain dataset setting. The results are shown in Table 2. Experimental results demonstrate that our method achieves significant outcomes across different domains. We achieved optimal performance on five datasets and second-best performance on three additional datasets. Notably, our approach demonstrated a strong lead on the Cars, EuroSAT, ISIC, and CropDisease datasets. This suggests that MSD also demonstrates strong generalization in cross-domain few-shot problems, reducing the impact of shortcut features during the fast adaptation phase.

### 5.2.3 AUGMENTED FEW-SHOT LEARNING PROBLEMS.

To further explore the enhancement of the model's learning capabilities initialized by MSD, we employed augmented tasks for testing. Specifically, during the meta-test phase, we augmented the support data for model fine-tuning and then classified the query data using the updated model. We report both the classification accuracy and the consistency of knowledge learned across different methods. Conv4 and ResNet12 were utilized to validate the generalization capabilities of MSD across varying scales.

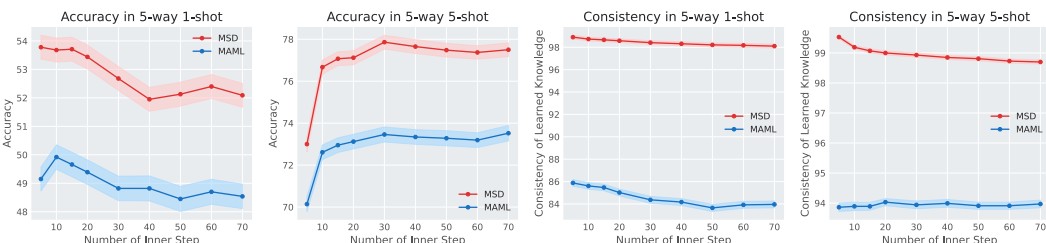

Figure 4: **The 5way-1shot and 5way-5shot classification accuracy and the consistency of learned knowledge with different numbers of inner steps** with 95% confidence interval, averaged over 2000 tasks

Table 5: **5way-1shot and 5way-5shot consistency of learned knowledge** in strong augmented few-shot classification task on Mini-ImageNet and Tiered-ImageNet (over 2000 tasks), using ResNet-12 as the backbone.

Table 6: **Ablation study on Mini-ImageNet.** All models are trained on the meta-train set of Mini-ImageNet.

| Methods | Mini-ImageNet | | Tiered-ImageNet | |
|---|---|---|---|---|
| | 1-Shot | 5-Shot | 1-Shot | 5-Shot |
| MAML | 85.88 | 94.03 | 84.93 | 93.87 |
| MSD + MAML | **98.58** | **99.00** | **99.70** | **99.80** |
| Unicorn-MAML | 87.55 | 94.60 | 86.67 | 95.41 |
| MSD + Unicorn-MAML | **99.91** | **99.92** | **99.94** | **99.96** |

| $\mathcal{A}ug$ | $\mathcal{L}_{\text{CK}}$ | Mini-ImageNet | |
|---|---|---|---|
| | | 1-shot | 5-shot |
| ✗ | ✗ | $64.43 \pm 0.46$ | $83.90 \pm 0.29$ |
| ✓ | ✗ | $64.31 \pm 0.48$ | $84.14 \pm 0.28$ |
| ✓ | ✓ | $\mathbf{65.41 \pm 0.47}$ | $\mathbf{84.88 \pm 0.29}$ |

**Augmented few-shot accuracy.** Table.3 presents the performance of Conv4 on the Mini-ImageNet dataset under varying levels of augmentation. MSD has an approximate 2% increase in classification accuracy on query data, irrespective of whether the perturbations are weak or strong. Table.4 demonstrates the performance of ResNet-12 under strong augmentation on both Mini-ImageNet and Tiered ImageNet datasets. It is evident that MSD confers greater improvements on models with larger capacities and contributes to a significant increase in accuracy for various tasks. This has further demonstrated the generalization of MSD.

**Consitency of learned knowledge.** Table.5 presents the consistency of knowledge acquired by the model variants for the same support data, as quantified by the similarity among the outputs of different model versions for the same query data, as shown in Eq.5. It is observed that both MAML and its variant, MAML-Unicorn, tend to learn inconsistency knowledge in both 5way-1shot and 5way-5shot scenarios. This implies that the model initialized by MAML and Unicorn-MAML is easily influenced by the different shortcut features produced by different augmentations, while our method achieves around 99% consistency in knowledge across both datasets for 5way-1shot and 5way-5shot problems. The result shows that our method significantly enhances the model's ability to learn consistently.

## 5.3 ABLATION STUDY

To further explore the effectiveness of MSD, we conducted some ablation studies on MSD. We focus on the affection of data augmentation and the number of inner steps.

**The impact of data augmentation and $\mathcal{L}_{\text{CK}}$** Table 5 illustrates the impact of data augmentation and $\mathcal{L}_{\text{CK}}$. The first row presents the results of MSD without data augmentation and $\mathcal{L}_{\text{CK}}$, which is equivalent to MAML. The second row shows the results of MSD without $\mathcal{L}_{\text{CK}}$, which is equivalent to MAML with augmentation. The third row displays the results of MSD. The result indicate that augmentation is not the primary factor in MSD's improvement. The main improvement is attributed to $\mathcal{L}_{\text{CK}}$, which enables the initialized model to learn consistently. This result further underscores the motivation to learn consistently.

**The impact of the inner step.** We further investigated the impact of different inner steps during the meta-test phase on the model's few-shot classification accuracy and precise learning capabilities.Fig.4 illustrates the impact of the number of inner steps during the meta-test phase on the

performance of the MSD algorithm. The results indicate that for any given number of inner steps, the models trained using MSD consistently outperformed those trained with MAML. Specifically, in the 5way-1shot and 5way-5shot tasks, MSD achieved an accuracy of approximately 7% and 4% higher than MAML, respectively. Concerning the consistency of the knowledge learned, there was a trend of decreasing consistency for both MAML and MSD as the number of inner steps increased. This suggests that an excessive number of inner steps during the meta-test phase may lead to the model learning shortcut features. However, MSD still maintained approximately 99% consistency in different settings of the inner step, which shows the robustness and generalization of MSD.

### 5.4 FURTHER ANALYSIS

**Compute consumption.** Compared to MAML, MSD achieves parity in algorithmic complexity by substituting different tasks with varied versions of the same task. Consequently, the computational overhead of MSD aligns with that of MAML.

**Visualization.** To further analyze the MSD on the learning capabilities of models, we visualized the models updated by augmented data as shown in Appendix Fig.5. Specifically, during the meta-test phase, we visualized models trained with Model-Agnostic Meta-Learning (MAML) and MSD. The model was first fine-tuned using augmented support data, with the number of inner steps set to 20. Then, query data was employed as the visualized data. Grad-CAM++ (Chattopadhay et al., 2018) was utilized to visualize the critical regions that the models focused on for understanding the query data. The visualizations reveal that the model trained with MAML tends to allocate more attention to the surrounding environment, potentially prioritizing it over the classified objects, while the model trained with MSD focuses more on the objects used for classification.

## 6 CONCLUSION

The tendency to learn shortcut features is the key challenge to few-shot learning. In our work, we observe that the model learned with higher consistency tends to be less influenced by the shortcut features. Building on this foundation, we introduce a meta-learning method named meta self-distillation(MSD). MSD updates the model respectively by utilizing different augmented views of support data in the inner loop, then maximizing the consistency of the outputs of the same query produced by different updated models. We evaluate MSD across three few-shot learning problems. MSD significantly enhances the performance of algorithms across various settings.

Learning to learn consistently is a new perspective for meta-learning. We believe our proposed algorithm represents a step forward in enhancing models' learning ability. Future research could extend such a framework to the domain of self-supervised learning and apply it to larger-scale models.

### REPRODUCIBILITY STATEMENT

The details of datasets, model architectures, hyper-parameters, and evaluation metrics are described in subsection 5.1 and Appendix A.1. Our code is attached to the Supplementary Material.

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

# A  APPENDIX / SUPPLEMENTAL MATERIAL

## A.1  HYPERPARAMETERS AND CODE ENVIRONMENT OF EXPERIMENT

**Hyperparameters.**

The hyperparameters has shown in the Table.7Table.8Table.9

**Calculate resources and Environment.**  Our experiment is conducted on NVIDIA A800 80GB PCIe and NVIDIA A100 40GB PCIe. We use Python version 3.10.14, PyTorch version 2.3.0, and CUDA toolkit 12.1 on A800 80GB, and use Python version 3.11.9, PyTorch version 2.3.0, and CUDA toolkit 11.8 on A100 40GB,

Table 7: Experimental Setup

| Parameter | Value |
|---|---|
| task batch Size | 4 |
| inner loop learning rate | 0.05 |
| outer loop learning rate | 0.001 |
| outer data points | 15 |
| outer loop learning rate decay | 1/10 every 10 epochs |
| coefficient $\gamma$ (Eq.9) | 1 |

Table 8: Augmentations for Strong-Augmented Few-Shot Scenario

| Augmentation | Parameters | Probability |
|---|---|---|
| Random Resize | (scale: 0.5–1) | - |
| Color Jitter | (0.8, 0.8, 0.8, 0.2) | 0.8 |
| Grayscale Conversion | - | 0.2 |
| Gaussian Blur | Expectation: 0.1, Variance: 2 | 0.5 |
| Random Horizontal Flip | - | 0.5 |

Table 9: Augmentations for Weak-Augmented Few-Shot Scenario

| Augmentation | Parameters | Probability |
|---|---|---|
| Center Crop | $84 \times 84$ | - |
| Color Jitter | (0.4, 0.4, 0.4, 0.1) | 0.8 |
| Grayscale Conversion | - | 0.2 |
| Gaussian Blur | Expectation: 0, Variance: 1 | 0.5 |
| Random Horizontal Flip | - | 0.5 |

## A.2  ALGORITHM

The specific algorithm flow of meta-self distillation is shown in Algo.2

## A.3  VISUALIZATION

We visualized the models updated by augmented data as shown in Appendix Fig.5. Specifically, during the meta-test phase, we visualized models trained with Model-Agnostic Meta-Learning (MAML) and MSD. The model was first fine-tuned using augmented support data, with the number of inner steps set to 20. Then, query data was employed as the visualized data. Grad-CAM++ (Chattopadhay et al., 2018) was utilized to visualize the critical regions that the models focused on for understanding the query data. The visualizations reveal that the model trained with MAML tends to allocate more attention to the surrounding environment, potentially prioritizing it over the classified objects, while the model trained with MSD focuses more on the objects used for classification.

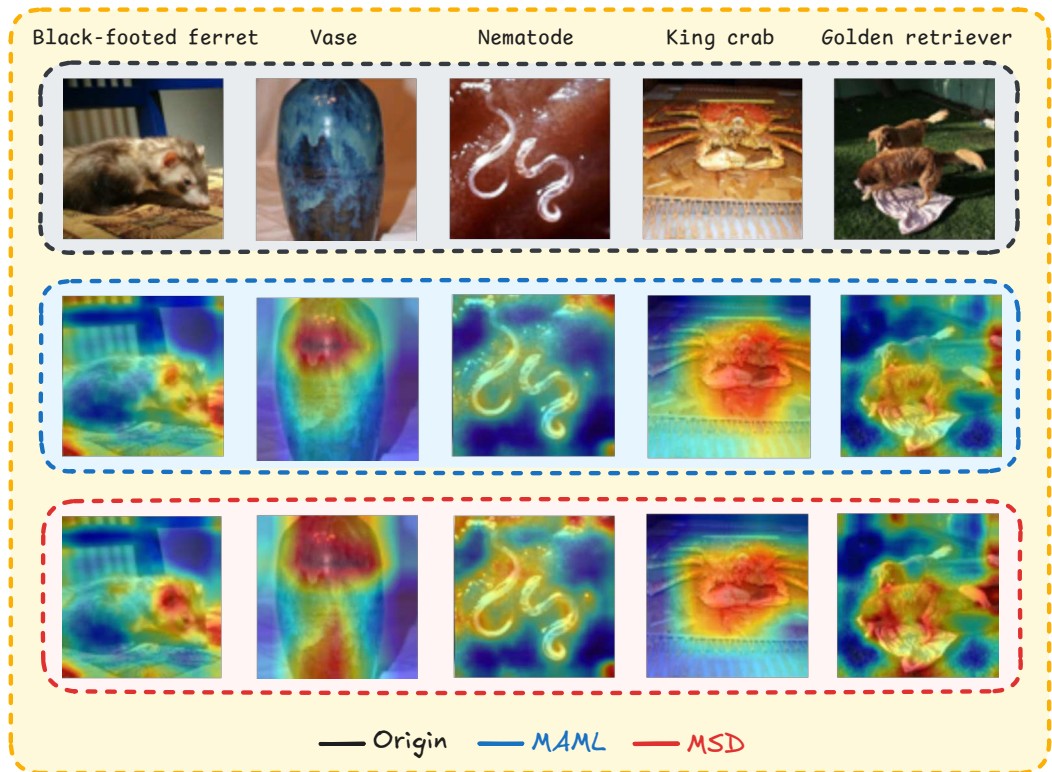

Figure 5: **The results of the visual analysis** on the test set of *Mini*ImageNet with MAML and MSD.

---

**Algorithm 2:** Meta Self-Distillation

---

**Given** the learned initialization $\theta^0$ pretrained on meta-train set
**for** $t \in \{1, \cdots, T\}$ **do**
    **Sample** task $\mathcal{T} = (\mathcal{S}, \mathcal{Q}) \sim \mathcal{D}_{\text{meta-train}}$
    **for** $i \in \{1, \cdots, n\}$ **do**
        **Random Augmented** the support data: Get $\mathcal{S}_i = \mathcal{A}ug(\mathcal{S})$
        **Update** $\theta^{t-1}$ by augmented support data $\mathcal{S}_i$: Get $\theta_i^{t-1} = \mathcal{U}^k(\theta^{t-1}, \mathcal{S}_i)$
        **Get the output** of the query data $x_q$ by $\theta_i^{t-1}$: Get $v_i = f_{\theta_i^{t-1}}(x_q)$
    **end**
    **Calculate** the outer loop loss: $\mathcal{L}_{\text{total}} = \mathcal{L}_{\text{cls}} + \gamma \cdot \mathcal{L}_{\text{CK}}$
    **Update** the parameters by outer loop loss: $\theta^t = \theta^{t-1} - \beta \cdot \nabla_{\theta^{t-1}} \mathcal{L}_{\text{total}}$
**end**
**Return** $\theta^T$

---

### A.4 COMPUTE COMSUMPTION

We counted the training time of MSD and MAML during the meta-train phase. Specifically, one epoch includes the optimization of 100 batches, where MAML uses 4 tasks for each batch for optimization, while MSD uses 1 task and enhances each batch 4 times for optimization. MSD has the same complexity as maml and thus has similar optimization times.

Table 10: **The training time of MSD and MAML during the meta-train phase**

| Time(Min) | Mini-ImageNet | Tiered-ImageNet |
|---|---|---|
| MAML (Finn et al., 2017) | 2.61 | 2.67 |
| MSD(Ours) | 2.76 | 2.85 |

