# OpenReview forum: "LEARN TO LEARN CONSISTENTLY"
_ICLR.cc/2025/Conference — ICLR 2025 Conference Withdrawn Submission_

### Official Review · Reviewer_a2G5 · 2024-11-01

**Soundness:** 2
**Presentation:** 3
**Contribution:** 2
**Rating:** 3
**Confidence:** 4

**Summary:**

The paper introduces a new framework for meta-learning in few-shot classification scenarios. The paper is motivated by how a neural network is susceptible to shortcut learning (learning easy features that often cause biases in learning) under the few-shot learning scenarios. The paper further goes on and suggests that a model that learns with higher consistency is less susceptible to shortcut learning. As such, the paper introduces a new meta-learning framework, named Meta Self-Distillation, that learns to learn consistently. The framework obtains multiple finetuned models, each with a different view of support data in inner-loop optimization. During outer-loop optimization, a meta-model (e.g., learnable initialization in MAML) is trained to maximize the consistency between multiple finetuned models.

**Strengths:**

- The paper is easy to follow.
- The paper introduces a simple idea of obtaining multiple models, each with different view/augmentation of the support set, in the inner-loop optimization, making it easier for meta-model to learn to maximize the consistency of learning
- The simple proposed idea leads to a performance improvement upon a baseline, MAML.

**Weaknesses:**

- In the paper, it says "models' weights are pre-trained on the meta-train set to initialize." I don't think this is how it's done for MAML. The performance comparison may not be fair due to this pretraining. Is this pretraining process done for all MAML-variant baselines in this paper's experiments?

- Another concern for experimental setting is the number of inner-loop steps. I could not find the number of inner loop steps used for quantitative comparisons, except for visualization. What is the number of inner loop steps used for MSD, and is this the same for all MAML variants?

- The paper seems to be missing the detail on the number of models (finetuned models) used in the inner-loop optimization. What is the exact number of resulting finetuned models in the inner-loop optimization? And, the paper should include the ablation study on the number of inner-loop models for better understanding & analysis of the proposed framework.

- There is a concern regarding technical novelty. The proposed framework seems to be a simple combination of MAML + multi-teacher distillation.
   - Learning from multiple teacher network, KDD 2017

- The paper should include a simple baseline with similar inference cost: Ensemble of MAML, where each MAML is learned separately with different augmentations (just like how it's done in inner-loop optimization of MSD). During inference, each MAML is finetuned with its corresponding augmentation and makes a final prediction after averaging. How does the performance compare?

- Related to the discussion above, the paper should include clear details on experimental settings.

- Why aren't MAML baselines compared in cross-domain few-shot classification? These experimental comparisons are needed to validate the claim that the proposed mitigates adverse effects of shortcut learning and gives better cross-domain few-shot classification performance.

- There are several works that try to address cross-domain few-shot classification via task-adaptive mechanisms. The experimental comparisons against these methods would strengthen the experimental results and the paper. Or, at the very least, the proposed method can be applied to these methods to show further improvements.
   - Learning to Balance: Bayesian Meta-Learning for Imbalanced and Out-of-distribution Tasks, ICLR 2020
   - Meta-Learning with Task-Adaptive Loss Function, ICCV 2021
   - Meta-Learning with Adaptive Hyperparameters, NeurIPS 2020 /  Learning to Learn Task-Adaptive Hyperparameters for Few-Shot Learning, TPAMI 2024
   - Learning to Forget for Meta-Learning, CVPR 2020 / Learning to Forget for Meta-Learning via Task-and-Layer-Wise Attenuation, TPAMI 2022
- Sharing a similar motivation with the paper, a paper below aims to tackle shortcut learning (learning spurious correlations) by causal intervention. The paper should provide discussions and experimental comparisons against this paper.
   - Interventional Few-Shot Learning, NeurIPS 2020

- The paper claims that learning with higher consistency leads to less shortcut learning. The paper further claims that the proposed method leads to higher consistency and thus less shortcut learning. However, Grad-CAM++ visualizations do not support the aforementioned claims much. While MSD is shown to give higher attention on the objects, there seems to be higher attention on background as well (Nematode and Golden retriever images). Furthermore, MAML seems to focus on the object to begin with. Better images would be the cases when there is no attention on the objects by MAML, while there is very high attention on the objects by MSD (refer to Figure 6 for better examples). Also the authors should show Grad-CAM++ visualization of other works (for example, ProtoMAML + papers mentioned above) that show similar performance to MSD but still susceptible to shortcut learning due to the lack of consistency.

**Questions:**

I have incorporated questions in the weakness section.

---

### Official Review · Reviewer_2gLT · 2024-11-02

**Soundness:** 3
**Presentation:** 2
**Contribution:** 2
**Rating:** 3
**Confidence:** 4

**Summary:**

The paper "Learn to Learn Consistently" addresses a critical issue in few-shot learning, where models tend to rely on biased shortcut features that limit generalization. To tackle this, the authors propose a novel meta-learning approach called Meta Self-Distillation (MSD), which enhances learning consistency across different augmented views of the same data. By promoting consistent knowledge retention, MSD encourages the model to focus on generalized rather than shortcut features, leading to improved performance on novel tasks. Extensive experiments demonstrate that MSD outperforms traditional methods in both standard and cross-domain few-shot learning scenarios, providing a robust solution to enhance generalization. This approach opens new pathways for meta-learning research, especially in areas like self-supervised learning and large-scale model adaptation.

**Strengths:**

1. According to reviewer’s investigation, the approach of "learning the representation consistency of the model across different tasks", is novel.
2. This paper accurately conveys its core idea.
3. The visualization approach of Figure5 is uncommon in meta-learning, but strongly supports the validity of the idea of this paper.
4. This paper provides a thorough survey of the field, which is reflected in the related work section.
5. The Figure in this paper is innovative, especially in Figure 1 and Figure 3.

**Weaknesses:**

# 1. Novelty
Three papers undermine the overall novelty of this paper. [1][2][3]

# 2. Presentation
There are many inconsistencies or omissions in the presentation of the article. While it is possible to guess the author's intent in context, these mistakes seriously detracts from the impact of the article. E.g.:
(1). The method in this paper is not actually related to distillation, while the authors consider their proposed approach to be based on self-distillation.
(2). Some arguments in the paper do not align with the referenced paper.
(3). Algorithm 1 is not be used in this paper. $C$ and  $C$ in Algorithm 1 are not consistent with $L_{cls}$ ​ and $L_{ck}$ in Algorithm 2.
(4). $S$ and $Q$ Equation 4. are reversed.
(5). $n$ and $f(.)$ are not defined.
(6). The paper contains excessive non-essential content, with Sections 2 and 3 taking up substantial space, which affects the paper's overall depth and substance.
(7). Unicorn-MAML is not cited.
(8). The authors should consider adding more qualifiers to the title, such as "few-shot image classification," to avoid unnecessary confusion.
(9) Some figures need to be draw more carefully.
# 3. Analysis
(1). According to the reviewer's understanding, a key characteristic of meta-learning is its ability to actively focus on shared representations across tasks under few-shot conditions, thereby avoiding intra-task overfitting (i.e., the "shortcut features" referred by the author). Study [4] shows that when the total number of tasks $n$ is sufficiently large, the model's generalization error will not be affected by intra-task overfitting. Study [5][6] also discuss a similar issue. However, the authors neither cite these papers nor discuss why the conclusions of this paper conflict with them.
(2). Adding a theoretical analysis discussing the impact of "consistency" on the generalization ability of meta-learning algorithms is necessary. References such as [7], [8], and [9] could be consulted for this purpose.
# 4. Experiments
(1). In the field of meta-learning, it is uncommon not to use the Omniglot dataset.
(2). The 20-way tasks should also be tested on other datasets to ensure a thorough comparison.
(3). Section 5.2.1 should use Table 1.(not Table 10).
(4). It is uncommon to use augmented few-shot learning problems to evaluate meta-learning performance. It would be worthwhile to explore how the gap between MSD and standard meta-learning algorithms changes under varying degrees of augmentation and under what conditions MSD has an advantage. The paper’s conclusion that “MSD has an approximate 2% increase in classification accuracy on query data, irrespective of whether the perturbations are weak or strong” makes this section seem somewhat redundant.
(5). This paper lacks accessible reference code.

# Reference
[1] Towards Reliable Neural Machine Translation with Consistency-Aware Meta-Learning
[2] Consistency-Guided Meta-learning for Bootstrapping Semi-supervised Medical Image Segmentation
[3] DAC-MR: Data Augmentation Consistency Based Meta-Regularization for Meta-Learning
[4] A Closer Look at the Training Strategy for Modern Meta-Learning
[5] Meta-Learning Requires Meta-Augmentation
[6] META-LEARNING WITHOUT MEMORIZATION
[7] Generalization Bounds for Meta-Learning via PAC-Bayes and Uniform Stability
[8] Information-Theoretic Generalization Bounds for Meta-Learning and Applications
[9] Algorithmic Stability and Meta-Learning

**Questions:**

The reviewer believes they have a general understanding of the entire content of the paper. If there are any misunderstandings, please clarify them in the rebuttal. Meanwhile, there are a few additional questions as follows.
1. According to the reviewer's understanding, a key characteristic of meta-learning is its ability to actively focus on shared representations across tasks under few-shot conditions, thereby avoiding intra-task overfitting (i.e., the "shortcut features" referred to by the authors). Study [4] shows that when the total number of tasks $m$ is sufficiently large, the model's generalization error is unaffected by intra-task overfitting. Study [5][6] also discuss a similar issue. However, the authors neither cite these two studies nor discuss why the conclusions of this paper conflict with them.
2. Based on generalization theory, what impact does consistency have on the generalization error of meta-learning algorithms?
3. Could the authors add more experiments to demonstrate that MSD also performs well in domains beyond image classification?

[4] A Closer Look at the Training Strategy for Modern Meta-Learning
[5] Meta-Learning Requires Meta-Augmentation
[6] META-LEARNING WITHOUT MEMORIZATION

---

### Official Review · Reviewer_Jxbm · 2024-11-03

**Soundness:** 3
**Presentation:** 2
**Contribution:** 2
**Rating:** 3
**Confidence:** 4

**Summary:**

This work highlights the positive connection between consistency and accuracy in MAML, finding that as a model's consistency learned from the task increases, the average accuracy in the predicting query data improves. The consistency refers to the similarity between output predictions of the same query but from models updated upon different augmented versions of the same set of support examples.

To this end, authors propose a multi-view learning method to learn consistently from different views, by accommodate MAML in a self-distillation framework, named Meta Self-Distillation (MSD).

In the classic inner-loop and outer-loop iterations, MSD learns to learn consistently from different views and shows improved FSL performance over other MAML variants.

**Strengths:**

This work proposes a patch to MAML to enhance its generalization. The idea of learning to learn consistently from different views in the inner- and outer-loop of MAML is novel in the scope of optimization-based FSL methods.

The motivation is well established.  The proposed multi-view learning in MAML is well formulated. Background knowledge is appropriately introduced. The presentation is fairly clear.

The proposed method shows improved performance over other MAML variants.

**Weaknesses:**

1) **The idea is novel but the contribution is incremental**

Data augmentation is widely used in FSL[1][2]. Seo etal proposed to augment samples in the image domain and regularized a model to learn consistent features in different feature levels, sharing an analogous idea. Of course, note that, they are quite different. The formulation in this work in the scope of optimization-based meta-learning is fancier.

2) **The presentation must be improved.**

States in Sec4.1 highly overlaps that in Sec1, and Fig2 is far away from where it is referenced. I recommend to move content in Sec4.1 into Sec1 to give a more direct and better motivation.

The overview of MSD (e.g. Fig3) is unkempt in a draft style, and can be improved. A professional illustration is prefered.

Tables are also far away from where they are referenced, which hinders reading and understanding.

As the contribution is the introduction of multi-view consistency alignment into MAML, the proposed MSD is essentially an optimization-based FSL method like MAML. So, I prefer to group metric-learning methods and optimization-based methods in different groups in one table (e.g. in Table 2-3). It helps in two aspects. Firstly, readers can clearly see that MSD is a MAML variant. Secondly, It clearly showcases the performance boost where compared both to metric-learning methods and optimization-based methods.

3) **Compare with other hallucination-based (or augmentation-based) methods**

It is recommend to compare with other hallucination-based methods that incorporate data augmentation for few-shot learning. Examples are [1][2].

4) **Others**

In Table2, the comparisons in cross-domain setting are mainly conducted against metric-learning methods. It's better to also consider comparisons to optimization-based methods, especially MAML and its variant, in the cross-domain setting, in order to straightforwardly demonstrate MSD's superiority.

Symbol $\bar{v}$ in EQ5 is not introduced. $\mathcal{F}_{sim}$ are introduced twice in both EQ5 and EQ7. $\mathcal{L}_{cls}$ is not properly introduced either.

In Tab3-4, MAML and Unicorn-MAML is not well referenced. It is also not recommended to ambiguously denote MSD as "MSD+MAML" or (MSD+Unicorn-MAML) in the tables. According to Sec4, MSD is not a play-and-plug module. It is a variant of MAML, like Unicorn-MAML. The self-distillation or augmentation, on the other hand, can be applied to other MAML variants.

5) **The paper is with many typos**

For example, inconsistent "maml" vs "MAML" in Sec 5.2.1.

"The results in Table 10" in Sec 5.2.1 should be Table1.

In Sec5.2 Line466, "inconsistency knowledge" vs "inconsistent knowledge".

In Sec 5.3 Line 477, "Table 5 illustrates the impact" vs "Table 6 ...".

In Sec4.1 Line 269, "alignment In various settings" vs ".. in various settings".



[1] Ni, R., Goldblum, M., Sharaf, A., Kong, K., & Goldstein, T. (2021, July). Data augmentation for meta-learning. In International Conference on Machine Learning (pp. 8152-8161). PMLR.

[2] Seo, J. W., Jung, H. G., & Lee, S. W. (2021). Self-augmentation: Generalizing deep networks to unseen classes for few-shot learning. Neural Networks, 138, 140-149.

**Questions:**

How to get the consistency? Is it the cosine similarity between embeddings of the same query? Or, is it the cosine similarity between predicted category distribution of the same query? (Fig3 shows it is the similarity between distributions.)

Is it the best practice to compare $v_i$ to $\bar{v}$ in EQ5?  How about $\max{v_i}$ or $\min{v_i}, rather than $\avg{vi}$?

Is MSD sensitive to the selection  data augmentation methods? Data augmentation has a nonnegligible  impact on unsupervised learning or contrastive learning.

---

### Official Review · Reviewer_748f · 2024-11-03

**Soundness:** 3
**Presentation:** 3
**Contribution:** 2
**Rating:** 5
**Confidence:** 5

**Summary:**

The authors propose a method to learn consistent feature representation for meta-learning, and they employ a common approach used in the self-supervised learning literature where multiple augmented views for a same training example are generated to enforce label consistency. To modify this approach for few-shot learning formulation, the authors update the model using few-shot support examples with different augmented views, and enforce label consistency across different models using a same query example. The proposed approach is validated on multiple few-shot classification datasets under comparison with various baseline few-shot learning algorithms.

**Strengths:**

- The motivation for the proposed method seems plausible and technically sound. With clear purpose of utilizing contrastive learning-based approach in the context of few-shot learning.

- The proposed learning method seems to be generally applicable to few-shot learning methods to avoid learning shortcut features.

- The authors validate their approach with various meta learning approaches, on well-known few-shot learning datasets including miniImageNet and TieredImageNet, under various shot settings and cross-domain experiments.

**Weaknesses:**

- Lack of validation under different baseline algorithms

  Since the proposed algorithm is based on MAML, which inspired a large number of variants that employ bi-level optimization scheme for few-shot learning, proposed method should be tested on more baseline algorithms. Can the proposed algorithm achieve performance gains across other MAML-based algorithms?


- Choice of data augmentations and hyperparameters

  Since the proposed algorithm involves various image augmentations on the support set data when meta-training, the choice of augmentations and the hyperparameters are crucial for improving the generalization and performance gains. By looking at Table 8 and 9 in the appendix, it seems that the proposed method requires a very specifically fine-tuned set of data augmentations to work properly. How sensitive is the proposed algorithm under different hyperparameter settings?


- Applications to datasets other than images

  MAML and other optimization-based few-shot learning algorithms are inherently "model-agnostic" where datasets of "any" modality can be utilized, and this is a strong advantage that makes MAML so versatile. Related to the previous question, proposed method seems to be "only" applicable to image-based few-shot learning problem, where the advantages of MAML are missing.


- Extension to use unlabeled query examples (semi-supervised settings)

  Since label consistency is enforced under different views (of support examples) similar to other self-supervised learning settings, can this method use unlabeled query data to do the same thing? Since there is an abundance of unlabeled examples, it seems to be a reasonable extension for research.

**Questions:**

Please refer to the questions in the weaknesses section. I am slightly positive on the simplicity of the idea proposed by the authors, but the proposed method seems to be only applicable to image data and its sensitivity to hyperparameter settings should be validated.

---

### Note · Authors · 2024-11-12

I have read and agree with the venue's withdrawal policy on behalf of myself and my co-authors.